# Revisiting the expressiveness of CNNs: a mathematical framework for feature extraction.

## Abstract

Over the past decade deep learning has revolutionized the field of computer vision, with convolutional neural network models proving to be very effective for image classification benchmarks. Given their widespread adoption, several works have attempted to analyze their expressiveness, and study the class of functions that they can realize. However, a fundamental theoretical questions remain answered: why can CNNs express discrete image classification functions that involve feature extraction? We address this question in this paper by introducing a novel mathematical model for image classification, based on feature extraction, that can be used to generate images resembling real-world datasets. We show that convolutional neural network classifiers can express a class of functions based on our simplified model of image classification datasets. In our proof, we construct piecewise linear functions that detect the presence of features, and show that they can be realized by a convolutional network.

## 1 Introduction

Over the past decade, convolutional neural network architectures have led to breakthroughs in a range of computer vision tasks, including image classification (12), object detection and semantic segmentation (21). Architectures such as AlexNet (6), VGG (22) and ResNet (13) have empirically shown that large convolutional neural networks can perform well on complex benchmarks such as ImageNet (3). While there has been theoretical research attempting to explaining their success in this context, fully understanding the underlying principles that contribute to their widespread adoption and effectiveness remains an open question.

From a mathematical perspective, two lines of work have emerged to theoretically explain the empirical success of neural networks. The first line focuses on their expressiveness, and examines the class of functions they can approximate. The second line investigates their learning capabilities, particularly their ability to learn these functions using stochastic gradient descent, and establishes bounds on their generalization error. The current work fits into the first category, and introduces a class of functions based on a simplified model of real-world vision data. A key result in this field is the universal approximation theorem for neural networks, which shows that they can approximate continuous functions with arbitrary precision (2), (14). While this result indicates that neural networks are suitable for regression problems, a key open problem is to establish an analogous result showing convolutional neural networks are suitable for discrete image classification tasks, that require the extraction of features from the input image.

It is well-known that convolutional neural networks with ReLu activations compute piecewise linear functions, and the complexity of these function classes has been studied in recent work (27), (16), (4), (17). Since their inception in the 1990s, it has also been empirically established that convolutional networks excel at feature extraction tasks (12). From a theoretical perspective, it is unclear why piecewise linear functions are effective for feature extraction. By answering this question, our key results bridge the gap between existing theoretical work analyzing the expressiveness of neural networks using piecewise linear functions, and empirical results on the success of convolutional neural networks for computer vision problems.

**Our contributions.** To address these problems, in this paper we present a rigorous mathematical framework that serves as a simplified model for image classification problems, which can be used to generate images

resembling real-world datasets. Our approach is based on the observation that an object corresponds to a set of constituent features. Intuitively, an object is present in an image precisely if one of the constituent features are present, though the location of these features can vary.

We analyze the expressiveness of convolutional network classifier, and construct networks can solve these image classification tasks perfectly, i.e. with zero error. In the proof, we construct piecewise linear functions that detect the presence of feature in an image, and show that these functions can be realized by convolutional neural networks. These piecewise linear functions are constructed by taking the sum of functions defined on patches in the input images; each function detects whether or not the patch contains a relevant feature. Our networks have several convolutional layers and one fully connected layers. The number of convolutional filters needed increases linearly with the complexity of the constituent features.

The paper is organized as follows. In Section 2, we introduce the convolutional neural networks that will be used and sketch the main results. In Section 3, we present our mathematical framework for image classification, with examples illustrating that it can be used to model real-world data. In Section 4, we present our main results on expressivness, showing that convolutional networks can realize this function class corresponding to image classification tasks, and outline the proof by constructing piecewise linear functions that extract features. In Appendix A, we perform an experimental analysis with our image classification framework, using features extracted from Fashion-MNIST. In Appendix B, we provide detailed proofs of the main results.

**Related work.**

*Approximation theory.* In the 1990s, it was shown that a neural network with a single one hidden layer can approximate any continuous function provided that its width is sufficiently large (2), (14). More recently, analogous results were established for deep neural networks; it was proven that fully connected ReLu networks with bounded width and unbounded depth can approximate continuous functions with arbitrary precision (8), (11), (15). Similar results were established for deep convolutional neural networks in (28). While these results give valuable insights, they do not explain why the class of continuous functions is suited for image classification tasks that involve feature extraction.

*Expressiveness of neural networks.* Another line of work analyzes the number of linear regions in the piecewise linear function that is computed by a neural network with ReLu activations (16), (19). Using combinatorial results, they derive lower and upper bounds for the maximal number of linear regions in a fully connected ReLu network with $L$ hidden layers and pre-specified widths (23), (1), (9), (10). These results show that deep fully connected networks can express functions with exponentially more linear regions than their shallower counterparts (18), (24). Analogous results for deep convolutional networks were established recently (27). Our work complements the above paper, by demonstrating that piecewise linear functions can also be use to extract features and solve image classification tasks.

## 2 Preliminaries

In this section we introduce notation, for convolutional neural networks and image classification tasks, that will be used throughout the paper.

### 2.1 Convolutional network architectures

**Definition 2.1.** We define the ReLu function $\sigma$, and the softmax function $\overline{\sigma}$ as follows. Here $\underline{x} = (x_1, \cdots, x_d)$ for some $d$.

$$\sigma(\underline{x})_i = \max(x_i, 0) \text{ for } \underline{x} \in \mathbb{R}^d$$

$$\overline{\sigma}(x)_i = \frac{e^{x_i}}{\sum_{i=1}^{k} e^{x_i}} \text{ for } \underline{x} \in \mathbb{R}^d \qquad \blacksquare$$

**Definition 2.2.** A fully connected layer with $n_1$ input neurons and $n_2$ output neurons consists of matrix $A \in \mathrm{Mat}_{n_1, n_2}(\mathbb{R})$ and biases $B \in \mathbb{R}^{n_2}$; we refer to the pair $W = (A, B)$ as the weights of the layer. We define

the map $\overline{\phi}_W : \mathbb{R}^{n_1} \to \mathbb{R}^{n_2}$ as follows (here $v \in \mathbb{R}^{n_1}$).

$$\overline{\phi}_W(v) = \sigma(Av + B) \qquad \blacksquare$$

**Definition 2.3.** A *convolutional filter* is a $k \times l$ matrix $w \in \mathrm{Mat}_{k,l}(\mathbb{R})$, which induces the following map. Here $\underline{x} \in \mathrm{Mat}_{m,n}(\mathbb{R})$ and $1 \le m' \le m - k + 1, 1 \le n' \le n - k + 1$.

$$\phi_w : \mathrm{Mat}_{m,n}(\mathbb{R}) \to \mathrm{Mat}_{m-k+1,n-k+1}(\mathbb{R})$$

$$\phi_w(\underline{x})_{m',n'} = \sum_{1 \le k' \le k, 1 \le l' \le l} w_{k',l'} \underline{x}_{k'+m'-1,l'+n'-1} \qquad \blacksquare$$

**Definition 2.4.** A *convolutional layer* consists of a set of convolutional filters $\underline{w} = (w_1, \cdots, w_f)$ and biases $\underline{b} = (b_1, \cdots, b_f)$. Here $w_i \in \mathrm{Mat}_{k \times l}(\mathbb{R})$ and $b_i \in \mathbb{R}$ for $1 \le i \le f$; we refer to the pair $(\underline{w}, \underline{b})$ as the weights of the convolutional layer. The induced map is as follows.

$$\overline{\phi^c_{\underline{w}}} : \mathrm{Mat}_{m \times n}(\mathbb{R}) \to \mathrm{Mat}_{m-k+1 \times n-k+1}(\mathbb{R})^{\oplus f}$$

$$\overline{\phi^c_{\underline{w}}}(\underline{x}) = (\sigma(\phi_{w_1}(\underline{x}) + b_1), \cdots, \sigma(\phi_{w_f}(\underline{x}) + b_f))$$

In the sum $\phi_{w_i}(\underline{x}) + b_i$, the bias term $b_i$ is added to each co-ordinate of the matrix $\phi_{w_i}(\underline{x})$. $\blacksquare$

**Definition 2.5.** A *flattening layer* is a linear isomorphism as follows, given by identifying $\mathrm{Mat}_{m',n'}(\mathbb{R})$ with $\mathbb{R}^{m'n'}$.

$$\phi_{fl} : \mathrm{Mat}_{m',n'}(\mathbb{R})^{\oplus f} \to \mathbb{R}^{m'n'f} \qquad \blacksquare$$

**Definition 2.6.** A convolutional neural network $\mathcal{N}$ consists of $L$ convolutional layers, a flattening layer, and $L'$ fully connected layers. Denote the weights of the $i$-th convolutional layer by $(\underline{w}_i, \underline{b}_i)$, and the weights of the $i$-th fully connected layer by $W_i = (A_i, B_i)$. The induced function $f_{\mathcal{N}}$ as follows. Here $m$ and $n$ denotes the height and width of the input image, and $l$ denotes the dimension of the output.

$$f_{\mathcal{N}} : \mathrm{Mat}_{m,n}(\mathbb{R}) \to \mathbb{R}^l$$
$$f_{\mathcal{N}}(\underline{x}) = \overline{\phi}_{W_{L'}} \circ \cdots \circ \overline{\phi}_{W_1} \circ \phi_{fl} \circ \overline{\phi^c}_{(\underline{w}_L, \underline{b}_L)} \circ \cdots \circ \overline{\phi^c}_{(\underline{w}_1, \underline{b}_1)}(\underline{x})$$
$$= (f^1_{\mathcal{N}}(\underline{x}), \cdots, f^l_{\mathcal{N}}(\underline{x}))$$

We denote by $\overline{f_{\mathcal{N}}}$ the classification function corresponding to the convolutional neural network $\mathcal{N}$.

$$\overline{f_{\mathcal{N}}} : \mathrm{Mat}_{m,n}(\mathbb{R}) \to [1, 2, \cdots, l]$$
$$\overline{f_{\mathcal{N}}}(\underline{x}) = \underset{1 \le i \le l}{\mathrm{argmax}} f^i_{\mathcal{N}}(\underline{x}) \qquad \blacksquare$$

## 2.2 Image classification

For image classification tasks, the input image is represented by rectangular matrices whose entries are scaled so their values between 0 and 1. Visually, the input image is divided into a rectangular grid, and the value of an entry in the matrix represents the color present in the corresponding portion of the rectangular grid (for instance, 0 could represent a white pixel, and 1 represents a black pixel).

**Definition 2.7.** Denote the input space as follows.

$$\mathcal{X}_{m,n} = \{\underline{x} = (x_{i,j}) \in \mathrm{Mat}_{m \times n}(\mathbb{R}) \mid 0 \le x_{i,j} \le 1\} \qquad \blacksquare$$

While color images are typically represented using multiple channels, for simplicity we only consider images which can be represented with a single channel (such as black-and-white images). We note however that it is straightforward to extend the results of this paper to the multi-channel setting.

We formalize the image classification problem below, using a pre-specified set of image labels $\mathcal{L}$ (such as "cat", "dog", etc). We restrict ourselves to a subset of the input space $\mathcal{X}_{m,n}$, consisting only of those matrices that correspond to one of the image labels. The objective is to construct an image classification map with zero error.

**Definition 2.8.** Let $\mathcal{L}$ be the finite set consisting of all image labels. For each image label $l \in \mathcal{L}$, let $\mathcal{X}_{m,n}^l$ denote the set of all input matrices that contain the image corresponding to $l$. We assume that the sets $\mathcal{X}_{m,n}^l, \mathcal{X}_{m,n}^{l'}$ are disjoint if $l \neq l'$. Denote by $\mathcal{X}_{m,n}^{\mathcal{L}}$ the set of all image matrices containing one of the images in $\mathcal{L}$.

$$\mathcal{X}_{m,n}^{\mathcal{L}} = \bigsqcup_{l \in \mathcal{L}} \mathcal{X}_{m,n}^l \qquad \blacksquare$$

**Definition 2.9.** An *image classification map* is a function $f : \mathcal{X}_{m,n}^{\mathcal{L}} \to \mathcal{L}$. We say that the map $f$ has zero error if the following holds.

$$\underline{x} \in \mathcal{X}_{m,n}^{\mathcal{L}} \Rightarrow f(\underline{x}) = l \qquad \blacksquare$$

We now give a sketch of the main results in this paper. For each label $l \in \mathcal{L}$, we formally define the set $\mathcal{X}_{m,n}^l \subset \mathcal{X}_{m,n}$, by specifying parameters that describe the features present in the corresponding image. We then construct a convolutional network classifier $f : \mathcal{X}_{m,n}^{\mathcal{L}} \to \mathcal{L}$ which has zero error, and present an upper bound on the number of neurons that it contains. The details of these constructions will be presented in the next section.

# 3 A mathematical framework for image classification

In this section, we present a rigorous mathematical framework for image classification problems and state our main results.

## 3.1 Image classification

We start with the observation that an image consists of a set of features that define it (4), (17), (25). For instance, a face consists of four typical features: a mouth, ears, eyes and nose. We proceed to rigorously define a feature.

For simplicity, our model stipulates that each feature can be characterized by a finite collection of fixed images. In the above example, a mouth would be defined by a set of distinct images, each of which resembles a human mouth. We introduce the notion of a "feature tile" to describe the constituent images.

**Definition 3.1.** Given a matrix $m \in \text{Mat}_{m,n}(\mathbb{R})$, define its *support* $\text{supp}(m)$ as follows.

$$\text{supp}(m) = \{(i,j)| \ 1 \leq i \leq m, 1 \leq j \leq n; m_{i,j} \neq 0\} \qquad \blacksquare$$

**Definition 3.2.** A **feature tile** $T$ with dimension $k \times l$ is a pair $T = (t, \epsilon)$ with $t \in \mathcal{X}_{k,l}$ and $\epsilon > 0$. $\blacksquare$

**Definition 3.3.** Given a feature tile $T = (t, \epsilon)$ with dimension $k \times l$ and an image $x \in \mathcal{X}_{k,l}$ define the quantity $\underline{t}(x)$ as follows.

$$\underline{t}(x) = \sum_{(i,j) \in \text{supp}(t)} |x_{i,j} - t_{i,j}| \qquad \blacksquare$$

The quantity $\underline{t}(x)$ is used to determine whether or not the image $x$ contains the tile $T$, with the parameter $\epsilon$ bounding the discrepancy between the two. The sum is taken over $\text{supp}(t)$ in the case where the feature that $T$ contains is not a full rectangle, but rather a subset of pixels inside a rectangle (i.e. the non-zero coordinates of $t$ contain the relevant feature). Below, we define the space of images $\mathcal{X}_{m,n}^T$ containing the tile $T$; the subscripts $[i+1, i+k] \times [j+1, j+l]$ specifies the region of the input image that contains it.

**Definition 3.4.** Given a feature tile $T = (t, \epsilon)$ with dimension $k \times l$, define $\mathcal{X}^T \subset \mathcal{X}_{k,l}$ and $\mathcal{X}_{m,n}^T \subset \mathcal{X}_{m,n}$ as follows. Given $\underline{x} \in \mathcal{X}_{m,n}$, below $\underline{x}_{[i+1,i+m],[j+1,j+n]}$ denotes the sub-matrix with rows indexed by $[i+1, \cdots, i+m]$ and columns indexed by $[j+1, \cdots, j+n]$.

$$\mathcal{X}^T = \{x \in \mathcal{X}_{k,l} \mid \underline{t}(x) \leq \epsilon\}$$
$$\mathcal{X}_{m,n}^T = \{\underline{x} \in \mathcal{X}_{m,n} \mid \exists \ i,j \text{ such that } \underline{x}_{[i+1,i+k],[j+1,j+l]} \in \mathcal{X}^T\} \qquad \blacksquare$$

We say that an input image contains an image $\mathcal{I}$ if it contains any of the constituent tiles corresponding to that image, so the corresponding subset of $\mathcal{X}_{m,n}$ is given by taking the union.

**Definition 3.5.** An **image** $\mathcal{I}$ is a set of **feature tiles**, $\mathcal{I} = \{T_1, T_2, \cdots, T_q\}$, with $T_i = (t_i, \epsilon_i)$ for $1 \leq i \leq q$. Note that the dimensions of the tiles $T_i$ need not be the same. Define $\mathcal{X}_{m,n}^{\mathcal{F}}$, the space of all images $\mathcal{X}_{m,n}$ containing the image $\mathcal{I}$, as follows.

$$\mathcal{X}_{m,n}^{\mathcal{F}} := \bigsqcup_{i=1}^{q} \mathcal{X}_{m,n}^{T_i}$$

Let $s(\mathcal{F}) = q$ be the number of feature tiles in the feature $\mathcal{F}$. Define the quantity $c(\mathcal{F})$ below.

$$c(\mathcal{F}) = \sum_{i=1}^{q} |\mathrm{supp}(t_i)| \qquad \blacksquare$$

Our image classification problem will be defined by a set of images. To avoid ambiguity, the corresponding set of image matrices will exclude any which contain multiple images (so the set $\mathcal{X}_{m,n}^{\mathcal{I}_j} \cap \mathcal{X}_{m,n}^{\mathcal{I}_{j'}}$ is excluded). In other words, we stipulate that any input matrix contains only one image.

**Definition 3.6.** An **image class** $\overline{\mathcal{I}}$ consists of a set of images $\overline{\mathcal{I}} = \{\mathcal{I}_1, \cdots, \mathcal{I}_l\}$. Define $\mathcal{X}_{m,n}^{\overline{\mathcal{I}}}$ to be the set of images which corresponds to exactly one of the images in $\overline{\mathcal{I}}$.

$$\mathcal{X}_{m,n}^{\overline{\mathcal{I}}} = \bigcup_{j=1}^{l} \mathcal{X}_{m,n}^{\mathcal{I}_j} - \bigcup_{1 \leq j < j' \leq l} \mathcal{X}_{m,n}^{\mathcal{I}_j} \cap \mathcal{X}_{m,n}^{\mathcal{I}_{j'}} \qquad \blacksquare$$

## 3.2 Examples

Here we use the above framework to model a real-world image classification task, with two labels: "cat" and "dog". Below we describe the defining features in more detail, and present examples from the image class that resemble real-world data.

Both of the image classes are defined by a single feature. For each feature, we specify two feature tiles of differing dimensions corresponding to the object. The "cat" (respectively, "dog") tiles are non-rectangular pictures that were extracted from real-world images of cats (resp. dogs). Both features are depicted below.

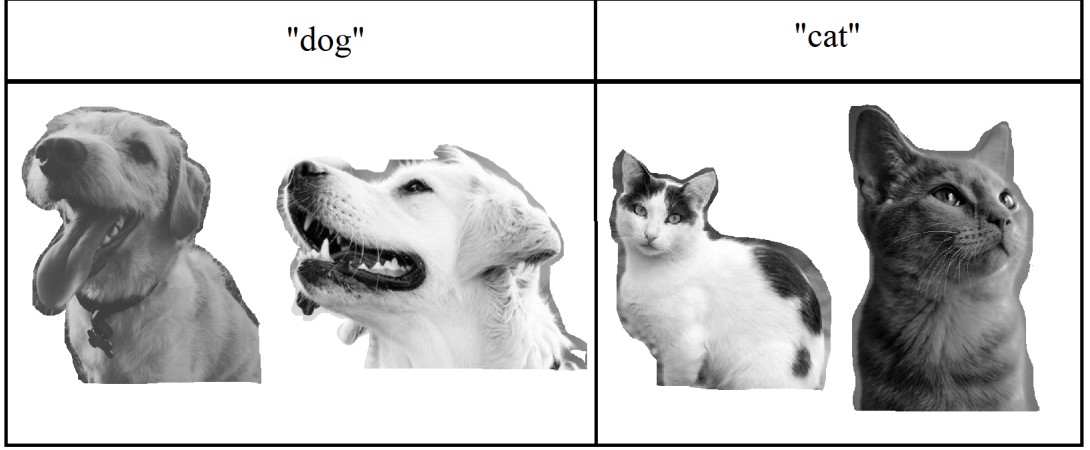

Figure 1: Features for "cat" and "dog"

Using the above features as building blocks, we present four examples from the the two image classes in the below figures. These examples are generated by superimposing one of the feature tiles above onto a background image. Note that our framework does not impose any restrictions on the background image. These resulting images are very realistic, and show that our framework can model a broad class of image classification tasks for a suitable choice of parameters.

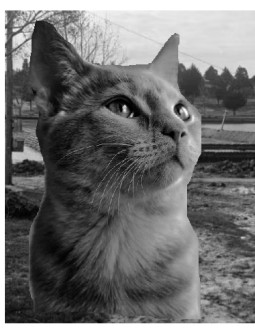 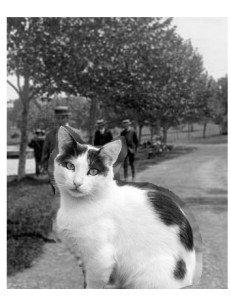 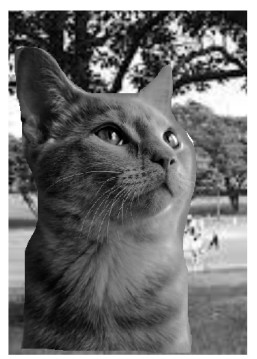 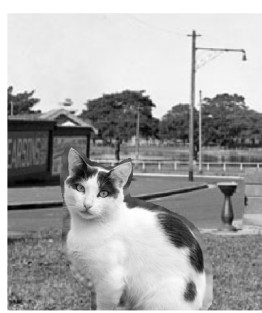

'cat'

Figure 2: Images from the "cat" class

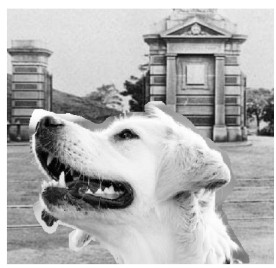 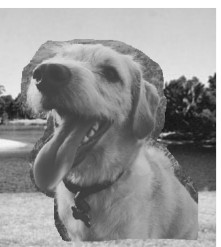 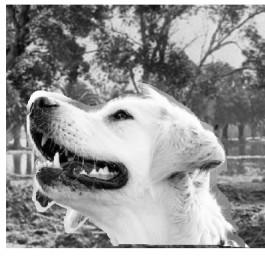 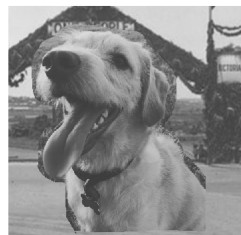

'dog'

Figure 3: Images from the "dog" class

While the above images show that our framework can generate interesting data in the case where each image consists of a single feature, more complex images can be generated by increasing the number of features. For instance, each of the two classes above could be divided into "eyes", "nose", "mouth" and "ears" features, resulting in a larger variety of images.

### 3.3 Main results

In order to solve the image classification problem presented in Section 3, we need to construct a convolutional network classifier that accurately predicts the labels of input matrices from $\mathcal{X}_{m,n}^{\overline{\mathcal{I}}}$. Our key result below constructs such a classifier.

Intuitively, we expect that each convolutional layer will be used to help identify the constituent feature tiles appearing in regions of the input image. The fully connected layers will be able to use the information extracted in the convolutional layers to solve the image classification task. See the discussion in Appendix A for more details about the number of parameters in the convolutional and fully connected layers of the network $\mathcal{N}[\overline{\mathcal{I}}]$, expressed in terms of the image class $\overline{\mathcal{I}}$.

**Theorem 1**. Let $\overline{\mathcal{I}} = \{\mathcal{I}_1, \cdots, \mathcal{I}_l\}$ be an image class, such that each image $\mathcal{I}_j$ contains at most $r$ features. There exists a network $\mathcal{N}[\overline{\mathcal{I}}]$ with one convolutional layer and one fully connected layers such that the induced classifier $\overline{f}_{\mathcal{N}[\overline{\mathcal{I}}]} : \mathcal{X}_{m,n}^{\overline{\mathcal{I}}} \to \overline{\mathcal{I}}$ has zero error.

It is widely understood that the success of deep convolutional networks is due to the principle of hierarchical compositionality (see (17)), whereby complex structures are obtained by combining simpler ones in a hierarchical fashion. Deeper layers in the network can recognize more complex features, building upon the

simpler features that were detected by earlier layers. Theorem 2 below, which is a variant of Theorem 1, highlights this concept.

**Theorem 2**. Let $\overline{\mathcal{I}} = \{\mathcal{I}_1, \cdots, \mathcal{I}_l\}$ be an image class, such that each image $\mathcal{I}_j$ has size less than $2^r$ for some $r$. There exists a network $\mathcal{C}[\overline{\mathcal{I}}]$ with $r$ convolutional layers and one fully connected layers such that the induced classifier $\overline{f}_{\mathcal{C}[\overline{\mathcal{I}}]} : \mathcal{X}_{m,n}^{\overline{\mathcal{I}}} \to \overline{\mathcal{I}}$ has zero error.

Convolutional neural networks compute piecewise linear functions, which leads us to the question of why these classes of functions can separate different image classes. The first key step in the proof is to construct piecewise linear function that can extract features from image, and detect whether or not an input matrix contains a given image. More precisely, we show the following.

**Proposition 1**. Given an image $\mathcal{I} \in \overline{\mathcal{I}}$, there exists a piecewise linear function $\phi_{\mathcal{I}}(\underline{x})$ such that the following statement holds.

$$\phi_{\mathcal{I}}(\underline{x}) : \mathcal{X}_{m,n} \to \mathbb{R}$$
$$\phi_{\mathcal{I}}(\underline{x}) > 0 \Leftrightarrow \underline{x} \in \mathcal{X}_{m,n}^{\mathcal{I}} \qquad \blacksquare$$

The second key step in the proof of Theorem 1 is to show that these piecewise linear functions can be realized using convolutional neural networks. The functions $\phi_{\mathcal{I}}(\underline{x})$ are constructed by defining functions that can detect the presence of features. We show that the latter functions can be realized by a single convolutional layer with multiple filters. We then use a fully connected layer to realize $\phi_{\mathcal{I}}(\underline{x})$ using these simpler functions. The proof gives us additional insight into the functions of individual neurons in the convolutional neural network.

In Appendix A, we conduct experiments with our image classification framework, focusing on special cases with features extracted from MNIST and Fashion-MNIST. We find that convolutional neural networks trained with stochastic gradient descent can achieve near-perfect accuracies in this context, provided that there is sufficient training data.

# 4 Proofs

In this section we outline the proofs of the main results; see Appendix B for details.

## 4.1 Proof of Proposition 1: constructing piecewise linear functions for feature extraction

The feature tile is the fundamental building block of an image class. As a first step towards the construction of $\phi_{\mathcal{I}}(\underline{x})$ needed for Proposition 1, we start by defining a piecewise linear function $\phi_T(\underline{x})$ for a given feature tile $T$, and show that an analogous statement holds in this setting.

**Definition 4.1.** We define $\mathcal{R}_{k,l}^{m,n}$, which indexes all sub-rectangles of size $k \times l$ inside a larger rectangle of size $m \times n$ as follows.
$$\mathcal{R}_{k,l}^{m,n} = \{(i,j) \mid i + k \leq m, j + l \leq n\} \qquad \blacksquare$$

**Definition 4.2.** Given a feature tile $T = (t, \epsilon)$ and an image $\underline{x} \in \mathcal{X}_{m,n}$, define $\phi_T(x)$ as follows.

$$\phi_T(\underline{x}) = \sum_{(i,j) \in \mathcal{R}_{k,l}^{m,n}} \max(0, \epsilon - \underline{t}(\underline{x}_{[i+1,i+k],[j+1,j+l]})) \qquad \blacksquare$$

**Lemma 4.3.** *The following inequality holds.*

$$\phi_T(\underline{x}) > 0 \Leftrightarrow \underline{x} \in \mathcal{X}_{m,n}^T \qquad \blacksquare$$

*Proof.* The inequality can be deduced as follows. Here $\underline{x} \in \mathcal{X}_{m,n}$.

$$\phi_T(\underline{x}) > 0 \Leftrightarrow$$
$$\underline{t}(\underline{x}_{[i+1,i+k],[j+1,j+l]})) < \epsilon \text{ for some } (i,j) \in \mathcal{R}_{k,l}^{m,n}$$
$$\Leftrightarrow \underline{x}_{[i+1,i+k],[j+1,j+l]} \in \mathcal{X}^T \text{ for some } (i,j) \in \mathcal{R}_{k,l}^{m,n}$$

By definition, this is equivalent to saying that $\underline{x} \in \mathcal{X}_{m,n}^{T}$. □

Using the above Lemma, now we are define the piecewise linear function $\phi_{\mathcal{I}}(\underline{x})$ with the desired property from Proposition 1.

**Definition 4.4.** Given an image $\mathcal{I} = \{T_1, \cdots, T_q\}$ and an input matrix $\underline{x} \in \mathcal{X}_{m,n}$, define $\phi_{\mathcal{I}}(\underline{x})$ as follows.

$$\phi_{\mathcal{I}}(\underline{x}) = \sum_{1 \le i \le q} \phi_{T_i}(\underline{x}) \qquad \blacksquare$$

*Proof of Proposition 1.* This can be deduced from Lemma 4.3 using the following argument. Note that since $\phi_{T_i}(\underline{x}) \ge 0$,

$$\phi_{\mathcal{I}}(\underline{x}) = \sum_{1 \le i \le q} \phi_{T_i}(\underline{x}) > 0$$

if and only if $\phi_{T_i}(\underline{x}) > 0$ for some $i$. This is true precisely when $\underline{x} \in \mathcal{X}^{T_i}$ for some $i$. In other words, it is true precisely when $\underline{x} \in \mathcal{X}_{m,n}^{\mathcal{I}}$. □

### 4.2 Proof of Theorem 1: realizing piecewise linear functions via convolutional networks

To prove Theorem 1, the key step is to construct convolutional neural networks that express the piecewise linear functions $\phi_{\mathcal{I}_j}(\underline{x})$. The image classification problem can then be solved using Proposition 1, which was proven in the previous section. Since these functions are built from the corresponding functions $\phi_T(\underline{x})$ for feature tiles, we start by showing that these can be expressed by a convolutional neural network. See Appendix B for complete proofs.

**Lemma 4.5.** *Let $T = (t, \epsilon)$ be a feature tile with dimension $k \times l$. There exists a convolutional neural network $\mathcal{N}[T]$ with one convolutional layer and one fully connected layer such that the following holds.*

$$f_{\mathcal{N}[T]}(\underline{x}) = \phi_T(\underline{x})$$

*The convolutional layer of $\mathcal{N}[T]$ has $4(|supp(t)| + 1)$ filters with $2 \times 2$ kernels, and the fully connected layer has less than $mn$ neurons.* ■

*Outline of proof.*

$$\phi_T(\underline{x}) = \sum_{(i,j) \in \mathcal{R}_{k,l}^{m,n}} \max(0, \epsilon - \underline{t}(\underline{x}_{[i+1,i+k],[j+1,j+l]}))$$

In the definition of $\phi_T(\underline{x})$ above, $\underline{t}(\underline{x}_{[i+1,i+k],[j+1,j+l]})$ is a sum of the terms $||x_{i',j'} - t_{u,v}||$.

$$||y - c|| = max(y - c, c - y) = \sigma(2y - 2c) - \sigma(y) + c \qquad \text{for } y, c \in \mathbb{R}$$

Using the above identity, $t(\underline{x}_{[i+1,i+k],[j+1,j+l]})$ can also be expressed as a linear combination of the quantities $\sigma(x_{i',j'})$ and $\sigma(2x_{i',j'} - 2t_{u,v})$, with a constant term. The latter quantities can be realized the outputs of a convolutional layer. The quantity $\phi_T(x)$ can be then realized by adding a fully connected layer. □

It is straightforward to extend the above Lemma to features, and construct convolutional neural networks that express the piecewise linear functions $\phi_{\mathcal{F}}(\underline{x})$ (see the Appendix B for a precise statement and proof). Now we are ready to construct convolutional neural networks that express the piecewise linear functions $\phi_{\mathcal{I}_j}(\underline{x})$, and outline the proof of Theorem 1.

*Outline of proof of Theorem 1.* For each image $\mathcal{I}_j$, denote the constituent feature tiles as follows. $\mathcal{I}_j = \{\mathcal{T}_1^j, \cdots, \mathcal{T}_{r_j}^j\}$. By Lemma 4.6 and Definition 4.4, there exists networks $\mathcal{N}'$ and $\mathcal{N}''$ such that the following holds.

$$f_{\mathcal{N}'}(\underline{x}) = [\phi_{\mathcal{T}_1^1}(\underline{x}), \cdots, \phi_{\mathcal{T}_{r_1}^1}(\underline{x}), \cdots, \phi_{\mathcal{T}_1^l}(\underline{x}), \cdots, \phi_{\mathcal{T}_{r_l}^l}(\underline{x})]$$

$$f_{\mathcal{N}''}[\phi_{\mathcal{T}_1^1}(\underline{x}), \cdots, \phi_{\mathcal{T}_{r_1}^1}(\underline{x}), \cdots, \phi_{\mathcal{T}_1^l}(\underline{x}), \cdots, \phi_{\mathcal{T}_{r_l}^l}(\underline{x})] = [\phi_{\mathcal{I}_1}(\underline{x}), \cdots, \phi_{\mathcal{I}_r}(\underline{x})]$$

By composing the two networks $\mathcal{N}'$ and $\mathcal{N}''$, and adding a softmax layer at the end, we obtain the desired network $\mathcal{N}[\overline{\mathcal{I}}]$, which has one convolutional layer and one fully connected layers. □

### 4.3 Proof of Theorem 2: hierarchical compositionality

To prove Theorem 2, the key step is to construct convolutional neural networks that express the piecewise linear functions $\phi_{\mathcal{I}_j}(\underline{x})$. The image classification problem can then be solved using Proposition 1, which was proven in the previous section. We build these functions using the principle of compositionality, and show that the functions $D_{\underline{t}}(\underline{x})$ defined below can be expressed by a deep convolutional neural network.

**Definition 4.6.** Given a matrix $t \in \mathrm{Mat}_{p,q}(\mathbb{R})$ and $\underline{x} \in \mathcal{X}_{m,n}$, define $D_{\underline{t}}(\underline{x}) \in \mathrm{Mat}_{m-p+1,n-q+1}(\mathbb{R})$ as follows. Here we using the notation from Definition 3.3, and assume $p < m$ and $q < n$.

$$
D_{\underline{t}}(\underline{x}) = \begin{pmatrix} \underline{t}(\underline{x}_{1:p,1:q}) & \underline{t}(\underline{x}_{1:p,2:q+1}) & \cdots \\ \underline{t}(\underline{x}_{2:p+1,1:q}) & \cdots & \cdots \\ \cdots & \cdots & \cdots \end{pmatrix}
$$

**Lemma 4.7.** *Given $a \in \mathbb{R}$, there exists a convolutional neural network $\mathcal{A}$ with two convolutional layers that has the following property. Below $D_{\underline{a}}(\underline{x})$ denotes the matrix from the above definition, with $a \in \mathrm{Mat}_{1,1}(\mathbb{R})$.*

$$
f_{\mathcal{A}}(\underline{x}) = D_{\underline{a}}(\underline{x})
$$

*Both convolutional layers have $1 \times 1$ kernels; the former has two filters, and the latter has one filter.*

*Proof.* We use the following identity.

$$
||x_{i,j} - a|| = max(x_{i,j} - a, a - x_{i,j}) = \sigma(2x_{i,j} - 2a) - \sigma(x_{i,j}) + a \qquad \text{for } y, c \in \mathbb{R}
$$

The first convolutional layer has two filters, and its weights and biases are chosen so that the corresponding outputs are $\sigma(2x_{i,j} - 2a)$ and $\sigma(x_{i,j})$. The second convolutional layer has one filter, and its weights are chosen so that the output is $||x_{i,j} - a||$ (using the above identity). $\qquad\square$

**Lemma 4.8.** *Let $t \in Mat_{2k,2k}(\mathbb{R})$ and $\underline{x} \in \mathcal{X}_{m,n}$ be matrices (as in Definition 3.2-3.4). We divide $t$ into four smaller matrices as follows.*

$$
t_{11} = t_{1:k,1:k}; t_{12} = t_{1:k,k+1:2k}
$$

$$
t_{21} = t_{k+1:2k,1:k}, t_{22} = t_{k+1:2k,k+1:2k}
$$

*There exists a convolutional layer with weights $\underline{w}(t)$, satisfying the following property.*

$$
\phi^c_{\underline{w}(t)}(D_{t_{11}}(\underline{x}), D_{t_{12}}(\underline{x}), D_{t_{21}}(\underline{x}), D_{t_{22}}(\underline{x})) = D_t(\underline{x})
$$

*The convolutional layer has one filter and $k \times k$ kernels.*

*Proof.* We use the following identity, which follows from the definitions.

$$
\underline{t}(\underline{x}_{i+1:i+2n,j+1:j+2n}) = \underline{t_{11}}(\underline{x}_{i+1:i+n,j+1:j+n}) + \underline{t_{12}}(\underline{x}_{i+1:i+n,j+n+1:j+2n})
$$
$$
+ \underline{t_{21}}(\underline{x}_{i+n+1:i+2n,j+1:j+n}) + \underline{t_{22}}(\underline{x}_{i+n+1:i+2n,j+n+1:j+2n})
$$

Let $\underline{w}(t) = (w_1, w_2, w_3, w_4)$, with the matrices $w_1, w_2, w_3, w_4 \in \mathrm{Mat}_{k,k}(\mathbb{R})$ defined below (here $E_{i,j} \in \mathrm{Mat}_{k,k}(\mathbb{R})$ denotes a matrix with a 1 in the $(i,j)$-th position, and zeroes elsewhere).

$$
w_1 = E_{11}, w_2 = E_{1,k}, w_3 = E_{k,1}, w_4 = E_{k,k}
$$

From the above expression for $\underline{t}(\underline{x}_{i+1:i+2n,j+1:j+2n})$, it follows that the map $\phi^c_{\underline{w}(t)}$ has the desired property. $\quad\square$

**Lemma 4.9.** *Let $T = (t, \epsilon)$ be a feature tile with dimension $k \times k$, where $k = 2^r$ for some $r \geq 1$, and let $\underline{x} \in \mathcal{X}_{m,n}$. There exists a convolutional neural network $\mathcal{N}[t]$ with $r + 1$ convolutional layers such that the following holds.*

$$
f_{\mathcal{N}[t]}(\underline{x}) = D_t(\underline{x})
$$

*The $(i+1)$-st convolutional layer of $\mathcal{N}[T]$ has $2^i \times 2^i$ kernels.* ∎

*Outline of proof.* We proceed by induction. The $r = 1$ case can be deduced from Lemma 4.7 as follows. As in Lemma 4.7, we construct the first convolutional layer so that the outputs are $\sigma(2x_{i,j} - 2a)$ and $\sigma(x_{i,j})$. We choose the weights of the second convolutional layer so that the resulting output is $D_t(\underline{x})$, by using the identities in Lemma 4.7 and Lemma 4.8.

For the inductive step, we argue as follows. We divide $t$ into four smaller matrices - $t_{11}, t_{12}, t_{21}$ and $t_{22}$ - as in Lemma 4.8. By the inductive hypothesis, there exists convolutional neural networks $\mathcal{N}[t_{ij}]$ such that $f_{\mathcal{N}}[t_{ij}](\underline{x}) = D_{t_{ij}}(\underline{x})$ (here $1 \leq i, j \leq 2$). By concatenating these four networks and adding the layer $\phi_{\underline{w}(t)}$ from Lemma 4.8, we obtain the desired convolutional network whose output is $D_t(\underline{x})$. $\square$

**Definition 4.10.** Let $T = (t, \epsilon)$ be a feature tile with dimension $k \times l$ where $k, l < 2^r$. Define the enlarged tile $T^{(r)} = (t^{(r)}, \epsilon)$ to be the feature tile where $t^{(r)} \in \text{Mat}_{2^r, 2^r}(\mathbb{R})$ is obtained by padding the matrix $t \in \text{Mat}_{k,l}(\mathbb{R})$ with zeroes (so that $t^{(r)}_{1:k,1:l} = t$). Given an image $\mathcal{F} = \{T_1, \cdots, T_q\}$, define the enlarged image $\mathcal{F}^{(r)}$ as follows:

$$\mathcal{F}^{(r)} = \{T_1^{(r)}, \cdots, T_q^{(r)}\}$$

Now we are ready to construct convolutional neural networks that express the piecewise linear functions $\phi_{\mathcal{I}_j}(\underline{x})$, and outline the proof of Theorem 2.

*Outline of proof of Theorem 2.* Our convolutional neural network will consist of a padding layer $p$, $r + 1$ convolutional layers, and two fully connected layers. We start with a padding layer $p$ which adds $2^r$ pixels on each of the four sides of the input.

For each image class $\mathcal{I}_j$, denote by $\{t_1^j, \cdots, t_{r_j}^j\}$ the tiles appearing in the features that constitute $\mathcal{I}^j$. Using Lemma 4.9, there exists a convolutional neural networks $\mathcal{N}'$ such that the following holds.

$$f_{\mathcal{N}'}(\underline{x}) = [D_{t_1^{1(r)}}(p(\underline{x})), \cdots, D_{t_{r_1}^{1(r)}}(p(\underline{x})), \cdots, D_{t_1^{l(r)}}(p(\underline{x})), \cdots, D_{t_{r_l}^{l(r)}}p(\underline{x})]$$

From Definition 4.6, it is easy to see that there exists a fully connected network $\mathcal{N}''$ with two layers such that the following holds.

$$f_{\mathcal{N}''}[D_{t_1^{1(r)}}(p(\underline{x})), \cdots, D_{t_{r_1}^{1(r)}}(p(\underline{x})), \cdots, D_{t_1^{l(r)}}(p(\underline{x})), \cdots, D_{t_{r_l}^{l(r)}}p(\underline{x})] = [\phi_{\mathcal{I}_1^{(r)}}(p(\underline{x})), \cdots, \phi_{\mathcal{I}_l^{(r)}}(p(\underline{x}))]$$
$$= [\phi_{\mathcal{I}_1}(\underline{x}), \cdots, \phi_{\mathcal{I}_l}(\underline{x})]$$

By composing the two networks $\mathcal{N}'$ and $\mathcal{N}''$, and adding a softmax layer at the end, we obtain the desired network $\mathcal{N}[\overline{\mathcal{I}}]$, which has $r$ convolutional layers and 2 fully connected layers. $\square$

### 4.4 Discussion and further directions

**Are convolutional layers needed for feature extraction?** One key advantage of convolutional layers is that weight sharing reduces the number of parameters that are stored, thus lowering the memory requirements. While it is possible to construct a network with the required properties in Theorem 1 using fully connected layers alone, the number of parameters needed would grow by an order of magnitude. In our image classification framework, one convolutional layer suffices as each features is represented by a discrete set of matrices. We also note our framework does not account for various subtleties of real-world vision data - for instance, CNNs are robust to wide range of distortions. These include scenarios where the image is rotated, and those where part of the image is partially occluded or distorted.

**How many convolutional layers are needed?** The success of convolutional neural networks for computer vision is predicated on the observation that deeper networks are better able to capture and represent more nuanced features. Deeper CNN architectures, such as VGG(22) and ResNet(13), perform better on real-world datasets (such as ImageNet(29)) than their shallower counterparts, like LeNet(12). The image classification task from Section 3 can be solved with a network that has a single convolutional layer (see Theorem 1). It would be interesting to generalize this, and construct a framework for image classification that can only be solved efficiently with deeper convolutional networks, where the features corresponds to a continuous spectrum of images.

**Stochastic gradient descent.** In practice, convolutional neural networks are trained with stochastic gradient descent on image classification tasks. This leads us to the question of whether the network constructed in Theorem 1 can be learned using stochastic gradient descent. As a step towards answering this, in Appendix A we conduct experiments using special cases of our image classification framework with features extracted from MNIST and Fashion-MNIST, and find that convolutional neural networks can achieve near-perfect accuracies in the large-data regime. It would be interesting to analyze this further from a theoretical perspective.

**Sparsity.** Empirically it has been observed that neural networks trained on computer vision datasets can be sparsified without a drop in accuracy (5), (7). Our theoretical results aligns well with these empirical observations. In the networks constructed in Theorem 1, most of the weights connecting the convolutional layer to the fully connected layer, are zero. This can be seen in the proof of Lemma 4.4; when $t(\underline{x}_{[i+1,i+k],[j+1,j+l]})$ is expressed as a linear combination of the quantities $\sigma(x_{i',j'})$ and $\sigma(2x_{i',j'} - 2t_{u,v})$, most of the coefficients are zero.

## 5 Conclusion

In this paper, we present a novel mathematical framework that can be used as a simplified model of real-world computer vision tasks. We focus on the image classification task, using convolutional neural network models that consist of convolutional layers and fully connected layers. In this context, we analyze the expressiveness of convolutional networks and show that they can solve image classification tasks, by constructing piecewise linear functions that extract features from the input image. We do not anticipate any negative societal impacts, as the present work is theoretical. Our work provides some insight into the theoretical underpinnings of computer vision, and we anticipate that our results can be generalized to provide a more detailed mathematical explanation as to why deep learning models can solve computer vision tasks.

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

# A    Appendix A: experimental results

In this section, we conduct extensive experiments on examples of our image classification frameworks that we construct, by extracting features from MNIST (12) and Fashion-MNIST (26). We describe the dataset generation process and the models used for experimentation in Section 1.1 and present our experimental findings in Section 1.2.

## A.1    Settings

**Dataset.** We generate image classes $\mathcal{I}$ as follows, with one class for each of the ten labels (in MNIST or Fashion-MNIST). Each image $\mathcal{I}$ consists of a single feature, and each feature consists of $k$ feature tiles. Each of the feature tiles are obtained by randomly choosing a sample from the training set of Fashion-MNIST with the corresponding label, and setting $t$ to be the $28 \times 28$ matrix obtained. To generate images from $\mathcal{X}^{\mathcal{I}}$, we start by randomly generating an input image with dimension $40 \times 40$. We then randomly select a rectangular subpatch with dimension $28 \times 28$, and replacing it with one of the $k$ feature tiles.

This process is illustrated in the below figure for Fashion-MNIST. One feature tile is randomly chosen for each of the ten classes in Fashion-MNIST. The two images directly underneath lie in $\mathcal{X}^{\mathcal{I}}$, and are obtained by pasting this feature tile in a random position onto a randomized background image.

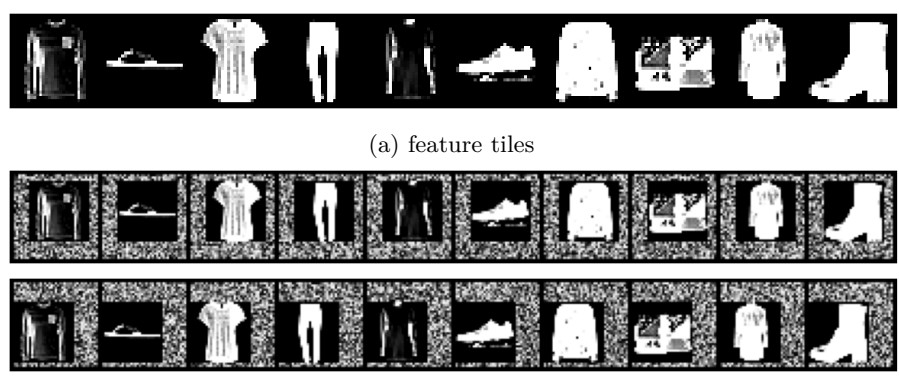

(a) feature tiles

(b) Images from $\mathcal{X}^{\mathcal{I}}$, for each of the ten classes

Figure 4: Images from $\mathcal{X}^{\mathcal{I}}$

**CNN architectures.** We use two convolutional neural network architectures for this image classification task. The first is a variant of LeNet (12) without the batchnorm layers. This network has a total of two convolutional layers, two maxpool layers, two fully connected layers and one output layer. Both convolutional layers have $5 \times 5$ kernels; the first has 6 convolutional filters, and the second has 12 convolutional filters. Both maxpool layers have kernels of size $2 \times 2$, with stride 2. The first fully connected layer has 120 neurons, and the second has 60 neurons.

The second is a simple convolutional neural network, which we refer to as the "simple ConvNet". It consists of a single convolutional layer followed by a fully connected layer and an output layer. The convolutional layer has $5 \times 5$ kernels, and 6 convolutional filters. The fully connected layer has 120 neurons.

**Training setup.** We use the same training setup for all experiments. The networks are trained using stochastic gradient descent with the Adam optimizer with learning rate set to 0.01, batch size 100, and a cross-entropy loss function. All layers are initialized using the Kaiming uniform initialization. We train the models for 10 epochs and all reported accuracies are averaged over five separate training runs. All experiments are conducted using PyTorch, with a single P100 GPU; see the supplementary sections for the code.

### A.2   Results

#### A.2.1   How does the size of the dataset affect the model's performance?

In this set of experiments, we generate datasets using the above procedure, by starting with $k$ random samples from MNIST (resp. Fashion-MNIST). We generate a training dataset with $n$ samples from each of the ten classes, for each $n \in \{50, 100, 150, 200, 300, 400\}$ and $k = 2$ . We train both of the above models on these datasets; the below graphs depict how the performance changes as the size of the dataset is increased.

| Dataset | Model | Dataset size | | | | | |
|---|---|---|---|---|---|---|---|
| | | 500 | 1000 | 1500 | 2000 | 3000 | 4000 |
| FashionMNIST | Modified LeNet | $0.92 \pm 0.06$ | $0.97 \pm 0.02$ | $0.99 \pm 0.01$ | $0.98 \pm 0.04$ | $1.00 \pm 0.00$ | $1.00 \pm 0.00$ |
| FashionMNIST | Simple Convnet | $0.83 \pm 0.06$ | $0.93 \pm 0.03$ | $0.98 \pm 0.00$ | $0.99 \pm 0.01$ | $0.99 \pm 0.02$ | $0.99 \pm 0.01$ |
| MNIST | Modified LeNet | $0.74 \pm 0.30$ | $0.98 \pm 0.01$ | $0.78 \pm 0.34$ | $0.89 \pm 0.23$ | $0.97 \pm 0.04$ | $0.99 \pm 0.01$ |
| MNIST | Simple Convnet | $0.48 \pm 0.17$ | $0.81 \pm 0.05$ | $0.74 \pm 0.30$ | $0.87 \pm 0.19$ | $0.94 \pm 0.11$ | $0.98 \pm 0.02$ |

Table 1: Final accuracy of models with varying dataset sizes

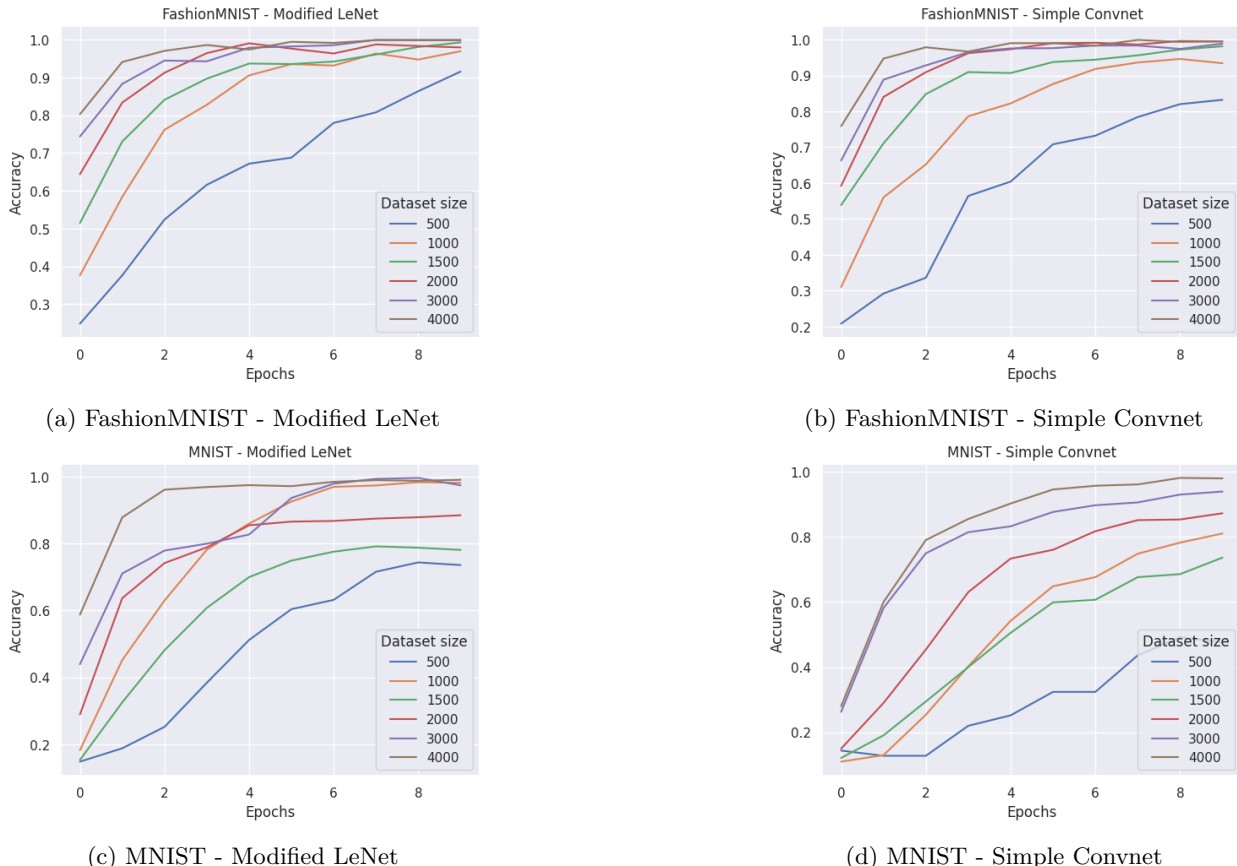

(a) FashionMNIST - Modified LeNet

(b) FashionMNIST - Simple Convnet

(c) MNIST - Modified LeNet

(d) MNIST - Simple Convnet

Figure 5: Model performance vs Dataset size

#### A.2.2   How does the complexity of the dataset affect the model's performance?

In this second set of experiments, we vary the complexity of the dataset (and not its size). We generate datasets using the above procedure by starting with $k$ random samples from MNIST (resp. Fashion-MNIST), for each $k \in [1, 2, 4, 8, 16, 32]$. We generate a training dataset with 320 samples from each of the ten classes.

We train both of the above models on these datasets; the below tables depict how the performance changes as the complexity of the dataset increases.

| Dataset | Model | k | | | | | |
| --- | --- | --- | --- | --- | --- | --- | --- |
| | | 1 | 2 | 4 | 8 | 16 | 32 |
| FashionMNIST | Modified LeNet | $1.00 \pm 0.00$ | $1.00 \pm 0.00$ | $0.95 \pm 0.01$ | $0.92 \pm 0.01$ | $0.80 \pm 0.04$ | $0.80 \pm 0.04$ |
| FashionMNIST | Simple Convnet | $1.00 \pm 0.00$ | $1.00 \pm 0.00$ | $0.95 \pm 0.01$ | $0.88 \pm 0.03$ | $0.76 \pm 0.02$ | $0.76 \pm 0.05$ |
| MNIST | Modified LeNet | $1.00 \pm 0.00$ | $1.00 \pm 0.00$ | $0.94 \pm 0.02$ | $0.83 \pm 0.05$ | $0.80 \pm 0.02$ | $0.84 \pm 0.04$ |
| MNIST | Simple Convnet | $0.99 \pm 0.01$ | $0.90 \pm 0.14$ | $0.88 \pm 0.07$ | $0.73 \pm 0.12$ | $0.63 \pm 0.18$ | $0.54 \pm 0.27$ |

Table 2: Final accuracy of models with varying k

### A.2.3 Discussion.

In all four settings, the above experiments indicate that convolutional neural networks trained with stochastic gradient descent can reach near-perfect accuracies on the image classification tasks that we analyze, provided that there is sufficient training data. Our results show that the amount of training data needed is heavily dependent on the complexity of the image classification task, which depends on the number of constituent features in each image class ($k$). In the case where $k = 2$, the models require approximately 300 images for each of the ten classes in order to reach near-perfect accuracies. As $k$ increases, the number of images needed per class also increases. On the other hand, a human classifier would be able solve these tasks with much fewer than 300 images per class. This indicates that while stochastic gradient descent performs well in the large-data regime, it would be interesting to develop alternative algorithms that are more effective when there is limited data.

## B  Appendix B: proofs from Section 4.1

We start by presenting a constructive proof of Lemma 4.4. This Lemma will play a key role in the proof of Theorem 1.

*Proof of Lemma 4.4.* Recall the definition of $\phi_T(\underline{x})$, as follows.

$$\phi_T(\underline{x}) = \sum_{(i,j) \in \mathcal{R}_{k,l}^{m,n}} \max\left(0, \epsilon - \underline{t}(\underline{x}_{[i+1,i+k],[j+1,j+l]})\right)$$

To simplify the notation, given $(i,j) \in \mathcal{R}_{k,l}^{m,n}$, define the following quantity.

$$\phi_{(i,j)}(\underline{x}) = \max\left(0, \epsilon - \underline{t}(\underline{x}_{[i+1,i+k],[j+1,j+l]})\right)$$

First we show that there exists a neural network $\mathcal{N}'[T]$ such that the following holds.

$$f_{\mathcal{N}'[T]} = [\phi_{(i,j)}(\underline{x})]_{(i,j) \in \mathcal{R}_{k,l}^{m,n}}$$

Define the four binary matrices as follows.

$$w_{11} = \begin{pmatrix} 1 & 0 \\ 0 & 0 \end{pmatrix}, \qquad w_{12} = \begin{pmatrix} 0 & 1 \\ 0 & 0 \end{pmatrix}$$

$$w_{21} = \begin{pmatrix} 0 & 0 \\ 1 & 0 \end{pmatrix}, \qquad w_{22} = \begin{pmatrix} 0 & 0 \\ 0 & 1 \end{pmatrix}$$

Define the vectors $\underline{w}_1 \in \mathrm{Mat}_2(\mathbb{R})^4$ and $\underline{b}_0, \underline{b}_1 \in \mathbb{R}^4$ as follows.

$$\underline{w}_1 = (w_{11}, w_{12}, w_{21}, w_{22})$$
$$\underline{b}_0 = (0,0,0,0); \qquad \underline{b}_1 = (1,1,1,1)$$

Define $S$, the set of all non-zero entries in the matrix $t$, as follows.

$$S = \{t_{u,v} \mid (u,v) \in \operatorname{supp}(t)\}$$

Let $d = |S|$. For convenience, we relabel the entries, so that $S = \{s_1, \cdots, s_d\}$.

We specify the weights $(\underline{w}, \underline{b})$ of the convolutional layer below. Here $\underline{w} \in \operatorname{Mat}_{2,2}(\mathbb{R})^{4(d+1)}$ denotes the convolutional filters, and $\underline{b} \in \mathbb{R}^{4(d+1)}$ denotes the biases.

$$\underline{w} = (\underline{w}_1, 2\underline{w}_1, \cdots, 2\underline{w}_1)$$
$$\underline{b} = (\underline{b}_0, -2s_1\underline{b}_1, \cdots, -2s_d\underline{b}_1)$$

Let $\underline{x}'$ be the image of the input image $\underline{x}$ under the map $\phi_{(\underline{w},\underline{b})}$ induced by the convolutional layer composed with a flattening layer.

$$\underline{x}' = \phi_f(\phi_{(\underline{w},\underline{b})}\underline{x}) \in \mathbb{R}^{4(d+1)(m-1)(n-1)}$$

By the above construction, it is clear that the quantities $\sigma(x_{i',j'})$ and $\sigma(2x_{i',j'} - 2s_l)$ appear as coordinates of $\underline{x}'$ (for all $1 \le i' \le m, 1 \le j' \le n$ and $1 \le l \le d$). We will need the following identity to obtain the quantities $\|x_{i',j'} - s_l\|$.

$$\begin{aligned}
\|y - c\| &= \max(y - c, c - y) \\
&= \max(2y - 2c, 0) + (c - y) \\
&= \sigma(2y - 2c) - \sigma(y) + c
\end{aligned}$$

Using the above equation and the definition, $\underline{t}(\underline{x}_{[i+1,i+k],[j+1,j+l]})$ can be expressed as a linear combination of the quantities $\sigma(x_{i',j'})$ and $\sigma(2x_{i',j'} - 2s_l)$, with a constant term. It follows that there exists a fully connected layer $\phi_{(\underline{w}',\underline{b}')}$ with the following property.

$$\phi_{(\underline{w}',\underline{b}')}\underline{x}' = [\phi_{(i,j)}(\underline{x})]_{(i,j) \in \mathcal{R}_{k,l}^{m,n}}$$

The desired neural network $\mathcal{N}[T]$ can be obtained by adding a fully connected layer with one output neuron to the network $\mathcal{N}'[T]$, with all weights equal to 1. The resulting network $\mathcal{N}(T)$ has $4(|\operatorname{supp}(t)| + 1)$ convolutional filters with dimension $2 \times 2$. The fully connected layer has $|\mathcal{R}_{k,l}^{m,n}|$ neurons, and $|\mathcal{R}_{k,l}^{m,n}| < mn$. The conclusion follows. $\qquad\square$

Next we extend Lemma 4.4, from functions corresponding to feature tiles to the analogous functions for features.

**Lemma 4.5** *Let $\mathcal{I}$ be an image, consisting of feature tiles $\{T_1, \cdots, T_q\}$. There exists a neural network $\mathcal{N}[\mathcal{I}]$ with one convolutional layer and one fully connected layer such that the following holds.*

$$f_{\mathcal{N}[\mathcal{I}]}(\underline{x}) = \phi_{\mathcal{I}}(\underline{x})$$

*Proof of Lemma 4.5.* We construct the networks $\mathcal{N}(T_1), \cdots, \mathcal{N}(T_q)$ from the above Lemma. Recalling the definition of $\phi_{\mathcal{F}}(\underline{x})$, it suffices to construct a network $\mathcal{N}(\mathcal{I})$ with the following property.

$$f_{\mathcal{N}(\mathcal{I})}(\underline{x}) = f_{\mathcal{N}(T_1)}(\underline{x}) + \cdots + f_{\mathcal{N}(T_q)}(\underline{x})$$

The convolutional (resp. fully connected) layer of $\mathcal{N}[\mathcal{I}]$ is obtained by concatenating the convolutional (resp. fully connected) layers of $\mathcal{N}(T_i)$, for $1 \le i \le q$. The weights are chosen so that the above identity holds.

$\qquad\square$

Theorem 1 now follows from Lemma 4.5 and Proposition 1, as described in Section 4.2.

