# OpenReview forum: "Revisiting the expressiveness of CNNs: a mathematical framework for feature extraction"
_TMLR — Rejected by TMLR_

### Review · Reviewer_bL8B · 2024-05-29

**Summary Of Contributions:**

This paper demonstrates in a principled mathematical way that a CNN with one convolution layer and one fully connected layer can be constructed in a manner that it can solve the classification problem by extracting features on the pixel level.

**Audience:**

Yes

**Claims And Evidence:**

No

**Requested Changes:**

I think it is important for the authors to tackle all four weaknesses I state above, or to convince me that I misunderstood.

Things of less importance is some syntactical errors like "novel mathematical model for image classification...that can be used to generate images" in the abstract.

**Strengths And Weaknesses:**

Strengths

The paper is trying to shed light in a domain that is quite important for the field and that is understudied.

Weaknesses

Overclaims/claims not supported by the results: The title "why do CNNs excel at feature extraction" and in abstract "why can they solve...that involves feature extraction" imply that this paper deals with the "why". We know that CNNs excel in image classification (from the literature) and we know that a CNN can be constructed in the manner the authors claim (from this paper). We don't know that this specific CNN will be learned in a practical CNN training scenario (let alone that this CNN is the reason for the CNN success). In my opinion, the above statements in this paper can only form a hypothesis, which is contradictory to the literature (see below).

Unclear definition on feature extraction: In the computer vision community "features" mostly mean vectors in a vector space. In my understanding in this paper features refere to pixel regions, which can be a bit hard to read for many people. Also my understanding from page 6 "...eyes, nose, mouth and ears features..." the features here represent well defined, countable objects that can fit in a bounding box and can be compared with your frame tile. Can you confirm or elaborate more if I misunderstood?

CNNs learn texture: We know from Geirhos et al [1] that CNNs in practice learn texture. Can your results include the findings of [1]? Can you define "features" in a manner that can include texture? Texture is way more challenging to define, given that it can be periodic and you can find an abstractly large number of regions that is close to your frame tile (in definition 3.3). Also by including texture as a feature, the assumption in page 1 "an object is present in an image precisely if all the features are present" stops making sense. If you decide not to include texture as a feature in your mathematical model, then could you discuss [1] and why you hypothesize something contradictory to it?

Oversimplifications: "an object is present in an image precisely if all the features are present": What about occlusions? what about 3d rotations that can hide some features? CNNs are robust to many of those things and the idea of just counting features cannot explain that. t(x) in Definition 3: the pixel level distance is not robust to the slightest of linear operations on an image, let alone complex real-world lighting differences. CNNs can be quite robust to that, can you argue that your mathematical framework can explain that?

[1] R. Geirhos, P. Rubisch, C. Michaelis, M. Bethge, F.A. Wichmann, and W. Brendel. "ImageNet-Trained CNNs are Biased Towards Texture; Increasing Shape Bias Improves Accuracy and Robustness", ICLR 2019

---

> ### Author Response · Authors · 2024-06-11
> **The weaknesses have been addressed.**
>
> We thank the reviewers for their comments. We have made the following changes to address these concerns (see the updated pdf of our submission).
>
> (1) The paper does not address feature learning, nor does it claim to do so. The introduction and abstract of the paper has been amended to clarify any confusion resulting from the usage of the phrase ‘why’ (eg. “we address the question: why can CNNs express discrete image classification functions that involve feature extraction?”). Readers will now understand that our paper focuses on the expressiveness of CNNs, and not about feature learning.
>
> (2) The definitions have been modified to clarify the notion of ‘features’; “framed tiles” have been changed to “feature tiles”. These “feature tiles'' represent the constituent features, and are defined rigorously. The image’s features do indeed correspond to well-defined objects that can fit in a bounding box, and can be compared with our feature tiles.
>
> (3) Our mathematical framework does not take into account more subtle phenomena - such as texture, rotations and occlusions. We will add a sentence in the “Further Directions” section (S4.4) listing these as questions that can be explored in future work. However, we note that there is no contradiction with Geirhos et al [1]. We study a simplified model of vision data; while it has similarities with real-world data, it cannot be used to model complex datasets like ImageNet. Therefore we cannot apply our results directly in the setting of Geirhos et al, which relies on experiments with ImageNet data. Our simplified model will be of interest to TMLR readers - although it is not a comprehensive theory that fully explains the success of CNNs for computer vision problems, it is a step forward from existing theoretical research on the expressiveness of CNNs.

---

> > ### Author Response · Authors · 2024-06-12
> >
> > As there's only a few days left in the response period, please let us know if you have further questions or comments. We would be happy to provide any further clarifications.

---

### Review · Reviewer_w3MR · 2024-05-31

**Summary Of Contributions:**

Assuming a data model where image classes are determined by the exact presence of a number of features (archetypal sub-images, e.g. of eyes / noses), the authors construct convolutional nets which solve the classification task, given ground-truth information about the data.

**Audience:**

No

**Broader Impact Concerns:**

/

**Claims And Evidence:**

No

**Requested Changes:**

If I am not misunderstanding something fundamental about this manuscript, I feel forced to assess that unfortunately it strongly over-claims, does not show anything about feature learning, and uses very classic concepts without showing something really new. Therefore the TMLR acceptance criteria of an interested audience and claim-evidence correspondence are not met, and cannot be met without fully rewriting the manuscript to change focus on a more specific and novel point.

If the authors can argue that my general points of critique (see weaknesses) are incorrect, I will be happy to go more into the details and draft a list of any minor requested changes.

**Strengths And Weaknesses:**

### Strengths
- clear presentation
- intuitive construction

### Weaknesses
- Seemingly exaggerated and false claim to explain feature learning, while all that seems to be shown is that in principle a CNN constructed using features as defined can solve the task. Nothing is shown about learning, or over-parameterized CNNs where feature vs. lazy learning would be relevant.
- Outdated result: the manuscript presents the state of research on feature learning in CNNs as if only the universal approximation theorems and results about exponential expressivity with depth where known, then provides itself a theorem which only shows that a certain function class can be implemented.
- The experiments seem disconnected from claims of the theoretical part. Especially it is not shown that the trained networks would learn a solution similar to the constructed one. Since the data model is simple, consisting of a few repeating archetypes with noise background, it is not surprising that the models learn the task well.
- Over-claims, for example: Title, abstract posing it is not known how CNNs can learn features, discussion generalizing strongly from little evidence, conclusion: "Our work sheds light on the theoretical underpinnings of deep learning, and we anticipate that it will lead to the design of more scientifically rigorous computer vision architectures in the future."

---

> ### Author Response · Authors · 2024-06-10
> **The weaknesses have been addressed, and we also request further clarifications.**
>
> We thank the reviewers for their comments. We have made the following changes to address these concerns (see the updated pdf of our submission), and request some further clarification.
>
> (1) The paper does not focus on feature learning, nor does it claim to do so. The introduction and abstract of the paper has been amended to emphasize that our papers focus on the expressiveness of CNNs, and not about feature learning. The concluding sentence has been rewritten to focus on the contributions made in this paper.
>
> (2) If the reviewer feels that the results in the current work are outdated, we request more specific references and details as to why this is the case. Can you provide references that are relevant, besides the cited papers on universal approximation theorems and exponential expressivity with depth? While it is well-known that sufficiently large neural networks can approximate continuous functions arbitrarily well, this does not a priori include the function class studied in the current paper. Our goal in this paper is to bridge this gap between theory and practice, as the function class we examine is a simplified model of real-world vision data, and this is of interest to TMLR’s readers.
>
> (3) The experimental section explores a related question, about whether CNNs can learn features within the framework that we present. See the discussion in Section 4.4 on stochastic gradient descent for more details. While we don’t answer this question theoretically, the experimental section suggests that CNNs can indeed learn features via stochastic gradient descent within our framework. We anticipate that this question can be addressed in future work. The data model is indeed quite simple, and the empirical results are not surprising.

---

> > ### Comment · Reviewer_w3MR · 2024-06-11
> > **Changing abstract and title has not solved the issues with the paper content**
> >
> > Thank you for the work on the response and for rewriting title and abstract.
> > However, this has not changed the content of the paper, and the underlying weaknesses have not been addressed. If I ask the question what previously unknown aspect of CNN expressivity the paper explains, and whether any such result is clearly presented and discussed, I have to answer both negatively. While the manuscript presents the construction of the classification task and the CNNs solving the task, but does not argue convincingly why this is novel and relevant. In particular, I do not see why the ability of CNNs to solve classification tasks as considered by the authors should not be covered by the existing results on the expressivity.
> >
> > If the authors believe that the way they are constructing the planted features in the CNN, or their toy-model image classification is instructive and can be used to gain novel insight, then these directions should be followed and the manuscript introduction and discussion be rewritten to clearly communicate and extensively discuss such insights. The experiments should be designed to support any such insight.
> > As written before, the result of the experiments is currently unsurprising and does not support any other relevant claim.
> > An extremely general discussion based on tangential evidence, as in the current manuscript should not pass the peer review.
> >
> > Similarly, the current version of the work does not seem to be embedded in an extensive exploration of the literature. Prompted to provide references I have listed a few below. Of course, these are entirely random and by no means representative for the state of the literature on expressiveness, feature learning, and decision boundaries. It is the work of the authors to extensively explore relevant adjacent literature.
> >
> > Cagnetta et al. (2023) How Deep Neural Networks Learn Compositional Data: The Random Hierarchy Model.
> > https://arxiv.org/abs/2307.02129
> >
> > Ilyas et al. (2019) Adversarial Examples Are Not Bugs, They Are Features.
> > https://proceedings.neurips.cc/paper/2019/hash/e2c420d928d4bf8ce0ff2ec19b371514-Abstract.html
> >
> > Daubechies et al. (2021) Nonlinear Approximation and (Deep) ReLU Networks.
> > https://link.springer.com/article/10.1007/s00365-021-09548-z#Sec17
> >
> > Balestriero & Baraniuk (2021) Mad Max: Affine Spline Insights Into Deep Learning
> > https://ieeexplore.ieee.org/abstract/document/9296823
> >
> > Keup & Helias (2022) Origami in N dimensions: How feed-forward networks manufacture linear separability.
> > https://arxiv.org/abs/2203.11355
> >
> > Zang et al. (2018) Tropical Geometry of Deep Neural Networks.
> > https://proceedings.mlr.press/v80/zhang18i.html
> >
> > Carlsson (2019) Geometry of Deep Convolutional Networks.
> > https://arxiv.org/abs/1905.08922v1

---

### Review · Reviewer_gEvo · 2024-06-03

**Summary Of Contributions:**

This paper examines the representation power of CNNs, particularly focusing on the feature extraction process and the piecewise linear functions induced by ReLU activation. The image classification problem is mathematically formulated, and it is proven that CNNs can achieve zero error in this task.

**Audience:**

No

**Broader Impact Concerns:**

I do not have any concerns.

**Claims And Evidence:**

Yes

**Requested Changes:**

Please address the concerns outlined in the Weaknesses section above.

**Strengths And Weaknesses:**

### Strength
- Many properties of neural networks remain unexplored, making theoretical analysis of neural networks, including CNNs, a highly relevant problem.

### Weaknesses
- First and foremost, this paper considers only training error and does not address generalization ability at all. The critical property we seek to understand is why and how such ML models can generalize well. It is already well-known that neural networks can achieve zero error if they are sufficiently large, given certain well-established conditions. Moreover, the analysis in this paper is limited to image classification by CNNs, whereas more general settings have been widely studied. Therefore, I do not believe TMLR readers will be interested in the findings of this paper.
- The presentation should be improved. Specifically, although this paper focuses on "features" of data such as images, features are never properly defined. This makes it difficult to understand which properties of the data or models are being mathematically analyzed and how these features are related.
- It would be interesting if empirical analysis regarding the extracted features and the resulting performance on real-world datasets were conducted.

---

> ### Author Response · Authors · 2024-06-10
> **The concerns in the weaknesses section have been addressed, and we request some further clarification.**
>
> We thank the reviewers for their comments. We have made the following changes to address these concerns (see the updated pdf of our submission), and request some further clarification.
>
> (1) The paper does not address the generalization ability of ML models. The introduction and abstract of the paper has been amended to emphasize that our papers focus on the expressiveness of CNNs, and not their generalization ability.
>
> (2) Could you please provide references for the statement that “more general settings have been widely studied” (than image classification by CNNs, which is the focus of the current work)? While it is well-known that sufficiently large neural networks can approximate continuous functions arbitrarily well, this does not a priori include the class of image classification functions. Our goal in this paper is to bridge this gap between theory and practice, by showing that CNNs can approximate a class of image classification that we define rigorously, which will be of interest to readers of TMLR.
>
> (3) The definitions have been modified; “framed tiles” have been changed to “feature tiles”. These “feature tiles” represent the constituent features, and are defined rigorously.
>
> (4) In the Appendix, we present an empirical analysis with CNNs, with a dataset constructed using a real-world example (Fashion MNIST).

---

> ### Author Response · Authors · 2024-06-12
>
> As there's only a few days left in the response period, please let us know if you have further questions or comments. We would be happy to provide any further clarifications.

---

### Author Response · Authors · 2024-06-15
**Summary of changes.**

We thank the reviewers for their comments, and summarize the changes below.


(1) The introduction and abstract of the paper has been amended to emphasize that our papers focus on the expressiveness of CNNs, and not about their generalization ability (or feature learning). Our claims on the expressiveness of CNNs are presented clearly, and supported by rigorous and detailed mathematical proofs.
(2) It is well-known that neural networks can approximate continuous functions arbitrarily well, but this does not a priori include the class of image classification functions. Our goal in this paper is to bridge this gap between theory and practice, by showing that CNNs can approximate a class of image classification functions that we define rigorously. This will be of interest to readers of TMLR, as it fills a gap in the literature on the expressiveness of CNNs.

---

### Decision · Action_Editor_q5a5 · 2024-07-08

**Recommendation:** Reject

**Comment:**

The authors propose a new mathematical model to understand the representation power of CNNs. While this manuscript presents an interesting perspective, the reviewers have raised concerns about the experiments and theories, such as the definition of a feature and potential oversimplification.

The reviewers have provided several constructive suggestions to enhance the manuscript:

1) Conducting additional experiments and analyses to demonstrate the generalization capabilities of the proposed method, particularly on larger datasets.

2) Per the suggestion of Reviewer bL8B, incorporating a more thorough discussion on the current understanding of CNNs will substantiate the claims made in the paper.

3) It is advised to extend the analysis to tasks beyond classification (suggested by Reviewer w3MR).

The authors are encouraged to revise the manuscript carefully according to the reviewers comments. Overall, this manuscript is not ready to be published right now.

**Audience:**

The contribution should be further claimed.

**Claims And Evidence:**

The experiments should be meticulously designed to better support the theories presented.

**Resubmission Of Major Revision:**

The authors may consider submitting a major revision at a later time.